# The Beneficial Effects of Soybean Proteins and Peptides on Chronic Diseases

**DOI:** 10.3390/nu15081811

**Published:** 2023-04-07

**Authors:** Sumei Hu, Caiyu Liu, Xinqi Liu

**Affiliations:** Beijing Advanced Innovation Center for Food Nutrition and Human Health, Beijing Engineering and Technology Research Center of Food Additives, National Soybean Processing Industry Technology Innovation Center, Beijing Technology and Business University, Beijing 100048, China; husumei@btbu.edu.cn (S.H.); 2130021016@st.btbu.edu.cn (C.L.)

**Keywords:** chronic diseases, soybean peptides, diabetes mellitus, obesity, cardiovascular diseases, cancer

## Abstract

With lifestyle changes, chronic diseases have become a public health problem worldwide, causing a huge burden on the global economy. Risk factors associated with chronic diseases mainly include abdominal obesity, insulin resistance, hypertension, dyslipidemia, elevated triglycerides, cancer, and other characteristics. Plant-sourced proteins have received more and more attention in the treatment and prevention of chronic diseases in recent years. Soybean is a low-cost, high-quality protein resource that contains 40% protein. Soybean peptides have been widely studied in the regulation of chronic diseases. In this review, the structure, function, absorption, and metabolism of soybean peptides are introduced briefly. The regulatory effects of soybean peptides on a few main chronic diseases were also reviewed, including obesity, diabetes mellitus, cardiovascular diseases (CVD), and cancer. We also addressed the shortcomings of functional research on soybean proteins and peptides in chronic diseases and the possible directions in the future.

## 1. Introduction

With the development of urbanization and the increase in sedentary habits, chronic diseases have become a worldwide public health problem. Chronic diseases are non-infectious diseases, but not some specific diseases, with complex etiology, slow development, and long duration [1]. Chronic diseases may be caused by lifestyle, environment, diet, or genetic factors. There were 28 million people who died of chronic diseases worldwide in 1990, and this number increased to 36 million in 2008 and 39 million in 2016 [1]. More than two-thirds of the deaths worldwide are believed to be caused by chronic diseases [1]. In 2019, seven of the top ten causes of death were due to non-communicable diseases, according to the World Health Organization (WHO) [2]. At present, chronic diseases mainly include obesity, diabetes, cardiovascular diseases (CVD), and cancer, and they are key causes of premature death in humans [3]. The pathogenesis of chronic diseases is relatively complex, such as imbalances in the protease network, which can lead to malfunctions in the cellular signal network [4].

The expression of matrix metalloproteinases (MMPs) and the imbalance of the phosphoryladylinositol 3-kinase (PI3K)/AKT/major target of rapamycin (mTOR) signaling pathway may both have an impact on the development of chronic diseases such as CVD, type 2 diabetes (T2D), and cancer [4,5].

As metabolic diseases, chronic diseases are associated with metabolic syndrome (MetS). MetS is not a disease itself but a comprehensive concept, representing factors that increase the risks of individual diseases (as shown in Figure 1) [6]. It has been reported that several components of MetS led to a significant increase in the risk of chronic diseases and that the risk factors for chronic diseases and the definition of MetS partially overlap [7]. Therefore, MetS may be a risk factor leading to the development of chronic diseases. Several diseases associated with both conditions have been identified, including obesity, T2D, CVD, and cancer [7].

Among these conditions (Figure 1), obesity is a major cause of MetS. Obesity may increase the susceptibility to insulin resistance, thus causing MetS. In 2005, the International Diabetes Federation recognized obesity as a necessary factor in the diagnosis of MetS [9], while Dr. Reaven objected to this and believed that insulin resistance might be the main cause of MetS [10]. MetS is an important risk factor for T2D and CVD (see Figure 2). With the increase in the number of patients with MetS, the number of patients with T2D and CVD also increased significantly [8].

Obesity includes metabolic healthy obesity (MHO) and metabolic unhealthy obesity (MUO) [11], while MHO is unstable and transient, and most patients with MHO will transition to MUO stage with the accumulation of fat [12]. Morgan Mongraw Chaffin and colleagues found that MHO would develop into MUO when it exceeded a certain baseline (odds ratio [OR]: 1.60; 95% confidence interval [CI]: 1.14 to 2.25), and the risk of CVD for those with MUO increased significantly [13]. Due to the fact that obesity is the cause of heart diseases and the relationship between obesity and CVD is mediated by MetS, MetS can be regarded as a sign of obesity accumulation at the exposure threshold [13].

On the other direction, obesity can cause MetS to develop into diabetes, while nonalcoholic fatty liver disease (NAFLD), the most common metabolic liver disease, is a continuum between them [14]. In the United States, about 30% of adults have NAFLD, and 20% of them are developed by individuals with obesity [15]. In a meta-analysis of 24 studies involving 35,599 type 2 diabetic patients, the prevalence of NAFLD in ordinary diabetes patients was 59.67% but increased to 77.87% in diabetes patients with obesity [16]. MetS leads to an increase in glucose levels in the body and excessive production of free fatty acids [17], and the degree of dysfunction of pancreatic beta cells was related to the severity of MetS [18]. When pancreatic beta cells exceed a certain metabolic capacity for a long time, their quality and function will be reduced, and their metabolic function will be damaged [17].

At present, the treatments for chronic diseases include physical exercise and diet therapy, as well as drugs for related symptoms. Bioactive peptides derived from food proteins have been recognized by the industry as improving health because of their low costs and low side effects. A variety of bioactive peptides from different foods have been reported for their bioactivities, including anti-hypertension, anti-diabetes, and anti-cancer activities [19,20,21]. Studies have shown that increased intake of plant proteins is associated with decreased risk of obesity, CVD, diabetes, cancer, and other symptoms [21]. Soybean peptides, as one of the popular bioactive ingredients derived from soybean proteins, have been utilized in many health aspects, such as anti-obesity, anti-diabetes, anti-CVD, anti-cancer, and antioxidant activities [22].

As a traditional plant, soybean has been planted in China for nearly 5000 years [23]. The United States introduced soybeans in 1965 [24] and has now become the world’s highest soybean production country, with the production volume reaching 45% of the world’s total output [25]. Later on, the cultivation of soybeans gradually developed in other countries, and it has become a popular cash crop in the world. Proteins are the most abundant nutrient in soybean, accounting for about 40% of all nutrients, and they are a very important plant source of dietary proteins [26]. Soybean proteins contain all twenty types of amino acids, including nine essential amino acids [27,28]. Its nutritional value is equivalent to that of animal protein, and therefore, it is considered to be a full-value protein [22].

Soybean peptides are derived by the hydrolysis of soybean proteins using different proteases, and they are mixtures of oligopeptides with 3–6 amino acids and molecular weights of 300–700 Da [29]. The physiological activities of soybean peptides are determined by the size of their relative molecular weights and the sequences of amino acids [30]. Their amino acid composition and proportion are the same as those in soybean proteins, but they are easier to absorb and more stable [31]. Soy proteins and peptides have been shown to be safe and non-toxic in the past, which is important for their further utilization [32].

In this paper, we reviewed the structure, function, absorption, and metabolic characteristics of soybean peptides. We also discussed their potential effects on the regulation and improvement of chronic diseases.

## 2. Structure and Metabolism of Soybean Peptides

Bioactive peptides are sequences consisting of 2–20 amino acids that can regulate or improve physiological functions and thus prevent or treat chronic diseases [33]. Enzymolysis is an effective method to produce functional peptides, but different durations of the hydrolysis process and different enzymes used in the process have a great impact on their functions and intensity [33]. Soybean peptides are also generated with the enzymolysis approach (Figure 3). The separation and identification of protein hydrolysates can help us understand the relationship between the structures and functions of some peptides. This is very important for improving the bioavailability of bioactive substances in the future.

### 2.1. Structure and Function

The structures of bioactive peptides are very important for their biofunctions. Understanding the structures of the peptides, including disulfide bond position, amino acid composition and sequence, molecular weight, hydrophobicity, and other structural characteristics, is very important for the design of new peptides and the improvement of their efficacy, bioavailability, physical, and chemical properties [34]. For instance, peptide segments with three or more disulfide bonds have higher stability [35].

Peptides with specific functions generally have certain structural characteristics. On the other hand, specific structures may contribute to specific functions. For instance, peptides with proline or hydroxyproline at the C or N ends have good angiotensin-converting enzyme inhibitory (ACE-I) activity [34]. The ability of peptides to bind to ACE [36] and the antioxidant activity of soybean protein hydrolysates [37] depend on the presence of hydrophobic amino acids at the C-terminus. The presence of three aromatic amino acids (Trp, Tyr, and Pre), hydrophilic and basic amino acids (His, Lys, and Pro), and hydrophobic amino acids (Leu, Phe, and Val) in the polypeptides enhances their antioxidant capacity [38,39]. Soybean β-conglycinin is one of the most abundant proteins in soybean, accounting for 24.7–45.3% (*w*/*w*) of total protein components [40]. Compared with normal soybean β-conglycinin, deglycation enhanced its antioxidant performance, and thus deglycosylation may be an innovative strategy to improve its performance [41]. The presence of Glu, His, Asp, Met, and Val can significantly enhance the antioxidant capacity of soybean peptides, even for those with large molecular weights [42]. Although it is generally believed that peptides with low molecular weights have strong antioxidant properties, the functional effectiveness of peptides is also related to the processing methods. For example, higher substrate concentration in the digestive process generates more small soybean peptides, which enhance the free radical scavenging activity of, α, α-diphenyl-β-picrylhydrazyl (DPPH)—an important indicator for the evaluation of antioxidant activity [43]. These studies indicate that the structures of soybean peptides play an important role in their functions. Lingrong Wen et al. identified 46 peptides with immunomodulatory activity, and most of them contained Try, Glu, and hydrophobic amino acid residues (Pro, Gly, Phe, Val, Leu) [29]. The binding of hydrophobic amino acids with Cys, Glu, Tyr, Asp, Trp, and Gln in the sequence is also important for immune regulatory activity [44]. Soybean peptides are mixtures of small peptides with different molecular weights. There is still a lot to do for the isolation and identification of small peptides with different functions.

The functions of soybean peptides have been studied in many aspects, including reducing blood fat [45], acting as an antioxidant [46], being anti-cancer [47], intestinal flora [48], being immunomodulatory [49], being anti-inflammatory [50], being anti-hypertensive [51], being anti-diabetic [52], and other physiological activities. However, there are still relatively few studies focusing on the effect of structures on the functions of peptides. Some peptides derived from soybean proteins have displayed many functions, yet there is not enough evidence to conclude that specific functions are associated with specific structures. Daliri and colleagues believed that peptides with multiple biological activities are better than those with a single biological activity. This is because peptides with multiple activities can play multiple beneficial roles at the same time [53]. Therefore, how to efficiently derive stable soybean peptides with specific structures and multiple functions merits further research.

### 2.2. Absorption and Metabolism

Bioactive peptides play physiological roles beyond their nutritional values. However, most bioactive peptides are in an encrypted state when they exist in the parent protein, where they cannot perform their functions [54]. Short peptides containing 2–20 amino acids have to be released from the parent proteins through enzymatic hydrolysis to activate their bioactivities [55]. Protein hydrolysates or short peptides have higher biological activities than complete proteins and/or amino acid mixtures [55]. After being digested in the digestive tract, some bioactive peptides can be absorbed through the intestinal tract to enter the blood circulation completely and play a role when they reach the corresponding target organs, while the others have local effects in the gastrointestinal tract [55].

Intestinal epithelial cells are a major obstacle to the absorption of any food ingredient. The activity of proteases on the surface of intestinal epithelial cells covered by microvilli may be the key factor that affects the stability and integrity of peptides as well as their operation and biological activities [56]. Differentiated Caco-2 cells have the morphology and function of mature intestinal epithelial cells and express brush border peptidase and transporters. They are useful in vitro models and can be used to study the stability, absorption, and transport of peptides [57]. To determine the effective utilization of a bioactive peptide, the differentiated Caco-2 cell lines were also used as an intestinal model to investigate the absorption of the peptides [58]. Although the Caco-2 cell model has been used as the best model in vitro for studying the absorption of different compounds for 35 years [57], its use in studying the absorption of food bioactive peptides is relatively new. One study examined the absorption of soybean β-conglycinin on Caco-2 cells after in vitro digestion mimicking the gastrointestinal tract and showed that 22 of 25 different peptide segments from the apical chamber samples were detected at the basolateral side of the transwell. This indicates that they could be absorbed by Caco-2 cells in vitro [41]. Gilda Aiello et al. found that three soybean peptides (IAVPGEVA, IAVPTGVA, and LPYP) were partially absorbed by Caco-2 cells in vitro and improved cholesterol metabolism in HepG2 cells by inhibiting the activity of 3-hydroxy-3-methylglutamate CoA reductase (HMGCoAR) [59].

In addition to absorption, Caco-2 cells are also used to evaluate the stability of bioactive peptides [60]. This is because Caco-2 cells can express brushborder peptidases and transporters, which can affect the stability and transport of peptide segments [60]. The transport of the peptide KPVAAP was detected for up to 60 min in the apical and basolateral sides of the transwell, indicating that it can be stably absorbed by Caco-2 cells [60]. After the soybean peptide segment, WGAPSL is digested in the gastrointestinal tract. The degradation of WGAPSL on both sides of apical and basal samples during the transportation of Caco-2 cells is determined. This indicates that WGAPSL can pass through the intestinal peptidase and mucus layer and be completely absorbed by the human body. This shows that WGAPSL has good stability after being digested in the gastrointestinal tract [61]. Due to the hydrolysis and absorption of the gastrointestinal tract, the biological activity and absorption stability of bioactive peptides in vivo may be different from those in vitro, while peptides need to be in active form to exert their biological activity in vivo. Therefore, we should pay attention to the biological stability and metabolic changes after the peptides are transported to the blood.

As peptidases exist widely in the body, including in the liver, kidney, blood, and other tissues, the mode, rate, and degree of how soybean peptides metabolize may be different in different target organs [62]. After entering the body, the peptide bonds within bioactive peptides are cut by the endopeptidases to form oligopeptides, and then the *N*-terminals or *C*-terminals are hydrolyzed by the exopeptidase (carboxypeptidase, aminopeptidase) into amino acids [62]. In addition, due to the high molecular weight, charged functional groups, and low lipophilicity of some peptide segments, they are easily blocked by the intestinal epithelial barrier. This results in a decrease in their bioavailability [63]. To improve their stability and bioavailability, various biochemical methods have been adopted, such as the substitution of unnatural amino acids and D-amino acids, cyclization, chemical modification (*N*-terminal and *C*-terminal), main chain modification, and nanoparticle formulation [64]. Understanding how bioactive peptides are metabolized and degraded by endogenous proteases is very important for functional food or drug design and the improvement of the metabolic stability of peptides [65].

The absorption and metabolism of soybean peptides are crucial to their biological effectiveness. It is very significant to investigate their absorption rate in vitro and the pathways they may have an impact on in vivo.

## 3. The Effects of Soybean Peptides on Chronic Diseases

Researchers are interested in exploring peptides and protein hydrolysates as active ingredients to prevent or treat chronic diseases. As a potent bioactive with an abundant source, soybean peptides have attracted a lot of attention, and the functions of soybean peptides have been widely investigated (Table 1). The functions of soybean proteins and peptides on chronic diseases, including anti-obesity, anti-diabetes, anti-CVD diseases, and anti-cancer activities, are of interest in this review (Table 1 and Figure 4).

### 3.1. The Effect of Soybean Peptides on Obesity

The rapid increase in the prevalence of obesity has become a major public health issue globally. According to the WHO, when a person’s BMI is ≥ 30 kg/m^2^, it is considered obesity, and BMI ≥ 25 kg/m^2^ is considered overweight [87,88]. In 2016, more than 39% of adults worldwide were overweight and about 13% were obese [89]. Abdominal obesity is closely related to chronic metabolic diseases such as T2D and CVD [87]. There are many treatments for obesity, such as drugs, surgery, and diets, with diet being the easiest and cheapest way to lose weight with no side effects. Protein has been widely used as a diet strategy to lose weight because of its high satiety effect [90]. It has long been shown that soybean protein has an anti-obesity effect, even better than whey and casein [91]. A random cross-balance experiment showed that fermented soybean had a better regulation effect on appetite regulating hormones (Acyl-ghrelin, insulin, and arginine) in obese girls than non-fermented soybean [92]. It showed a higher insulin-stimulating effect, which may be because the fermentation process can accelerate the degradation rate of protein and increase the bioavailability of short peptides [92]. Current studies show that soybean protein components can play a certain anti-obesity role.

β-conglycinin accounts for about 20% of the total soybean proteins, making it an important component for the beneficial effects of soybean proteins. Studies have found that β-conglycinin can reduce serum triglyceride and cholesterol levels, and thus it may have an anti-obesity effect [69]. The diet containing soybean protein reduced weight and fat tissue accumulation in C57BL/6 mice, which showed that β-conglycinin played an anti-obesity role [66]. After a single intake of β-conglycinin, both fibroblast growth factor 21 gene (FGF21) expression in the liver and FGF21 in the circulating body of the mice increased significantly [67]. In addition, β-conglycinin feeding for up to 9 weeks kept FGF21 levels in the liver and circulating FGF21 at a certain level, thereby reducing weight gain associated with a high-fat diet and thus ameliorating obesity [67]. Similarly, conglycinin peptide also reduced the liver lipase activity in obese rats, thereby reducing the abdominal fat accumulation and lipid contents. This indicates that it may have an anti-obesity effect [68]. As shown above, β-conglycinin, as the main component of soybean proteins, may be a potential compound for the treatment of obesity. In the future, the anti-obesity effect of β-conglycinin and the underlying mechanism will need further investigation. It may be a good choice for patients with obesity to lose weight.

Soybean proteins were also involved in the regulation of the gastrointestinal microbiome and bile acid homeostasis [21,71,72,93]. The diverse microbiota plays a key role in the development of obesity, and their interactions (via signaling molecules/communication) can be maintained through diet/supplements [94]. The maintenance of the microbiota through dietary strategies may be of great importance in the treatment of chronic diseases. Consumption of soybean proteins improved the intestinal microbiota, increased the diversity of intestinal microbes, and improved the transmission of bile acid metabolism signals [71]. Muhammad Umair Ijaz et al. found that in high-fat diet-fed C57BL/6J mice, soybean protein supplementation increased the ratio of Firmicutes to Bacteriodetes, improved the composition of intestinal microorganisms, and reduced the accumulation of serum triglycerides [72]. The changes in intestinal microorganisms have a big impact on the risk of chronic diseases. Soybean proteins can improve the microbial diversity of the gastrointestinal tract, regulate fat synthesis, and thus have an anti-obesity effect. Soybean proteins can also reduce adipocyte hypertrophy, the concentrations of free fatty acids, and the accumulation of triglycerides in the liver after high-fat diet intake [21]. Reza Hakkak and colleagues reported that in obese Zucker rats, soybean protein isolate (SPI) feeding for 8 weeks reduced the denaturation of fat in the liver and decreased the levels of aspartate aminotransferase (AST) and alanine aminotransferase (ALT) in serum, which were beneficial to bile acid homeostasis, and the effect was even better than that of casein [70]. Soybean protein consumption also reduced the production of fat in the liver of obese OLETF rats and reduced their cholesterol levels [71]. In another study, SPI reduced the weight and improved the body fat of rats significantly, and this may be related to the reduction of perirenal fat [93].

Previous studies have focused on the effect of soybean proteins on serum hormone levels, cholesterol metabolism, and gut microbiota, through which soybean proteins showed their anti-obesity effect [69,72]. However, there are relatively few studies investigating the anti-obesity effect of soybean peptides and the underlying mechanisms. As soybean peptides have been released from the parent soybean protein, they have a big potential to show anti-obesity effects. Further research is needed to unravel the anti-obesity effect of soybean peptides.

### 3.2. The Effect of Soybean Peptides on Diabetes Mellitus

At present, diabetes has become a public health problem worldwide, being the seventh leading cause of death globally [95,96] and causing a huge burden for the families of patients and the world economy. According to the WHO and the International Diabetes Federation, the prevalence of diabetes is increasing year by year. Diabetes is a systemic metabolic disease caused by abnormal blood glucose levels, and it is one of the fastest-growing and most common chronic diseases in the world [97]. There are two main types of diabetes, type 1 diabetes (T1D) and T2D [97]. T1D is mainly an early-onset autoimmune disease caused by genetic factors, which occurs in childhood mainly due to the reduction of pancreatic beta cells [97]. T2D is a late-onset non-autoimmune disease caused by environmental factors and characterized by beta cell dysfunction of the pancreas and insulin resistance [97]. The incidence of T2D is as high as 90%, and it is associated with a high mortality rate and high medical costs, as well as various complications, including retinopathy, kidney disease, microvascular complications, and nerve damage [97]. It is of great importance to find effective treatments for diabetes. Of many options, diet intervention is easy and cheap. Food-derived bioactive substances could be a good strategy to control blood glucose as a diet intervention strategy.

Soybean proteins are also high-quality proteins, but with different amino acid patterns, so they stimulate insulin secretion to different degrees than caseins [68]. Not only soybean proteins, but bioactives with amino acid patterns similar to soybean proteins also improved insulin sensitivity [68]. Soybean proteins are known as a hypoglycemic functional food as they contain specific amino acids, including Leu, Arg, Ala, Phe, Ile, Lys, and Met, which can stimulate insulin secretion and act as a trypsin inhibitor [98].

The relationship between the consumption of legumes and soybean products and the incidence of T2D was analyzed by Jun Tang and colleagues, and the results showed that specific soybean components, including soybean isoflavones and soybean proteins, were negatively related to the risk of T2D [99]. An increase of 10 g of soybean protein per day can reduce the risk of T2D by 9% [99]. The intake of soybean proteins was negatively correlated with the risk of diabetes in women in a dose-dependent manner but not in men [100]. Judit Konya et al. administered soybean proteins with or without soybean isoflavones to 60 diabetic patients aged 45–80 years and found that soybean proteins without isoflavones had a certain intrinsic activity in controlling blood glucose. This demonstrates the hypoglycemic effect of soybean proteins [75]. The structure, activity, and mechanism of plant compounds with therapeutic and ameliorative effects on T2D have been reviewed, and it showed that soybean proteins and bioactives play a certain role in the regulation of blood glucose [101]. Soybean proteins are rich in glycine and arginine, and their amino acid pattern is conducive to insulin sensitivity and glucose utilization [100]. These studies showed that the structural pattern of soybean proteins and their special amino acids can improve the sensitivity of pancreatic β cells and stimulate insulin secretion. Therefore, some soybean proteins and their derived bioactives may become effective substances for regulating blood glucose.

Dipeptidyl peptidase-IV (DPP-IV), α-glucosidase, and α-amylase are key enzymes that directly regulate blood glucose, and inhibition of these enzymes is an effective strategy for the treatment of T2D [102]. Gonzalez-Montoya et al. derived soybean peptides with different molecular weights after in vitro simulated gastrointestinal digestion. They found that peptides at 5–10 kDa inhibited DPP-IV, α-amylase, and α-glucoside, and further separation of these peptides yielded four components, three of which contained most polypeptides with encrypted dipeptide and tripeptide amino acid sequences, whose structure may be the main reason for their diabetes-inhibiting effects [52]. Another study showed that the soybean protein hydrolysates inhibited α-glucosidase activity and the peptides with molecular weight < 5 kDa and amino acid sequences Glu-Ala-Lys and Gly-Ser-Arg showed the strongest inhibitory effect [73]. Val-His-Val-Val (VHVV) is also a short peptide isolated from soybean protein hydrolysates. VHVV could restore the viability of H9c2 cells at high glucose conditions, and 10 μg/mL VHVV reduced the number of H9c2 cells undergoing apoptosis and postprandial blood glucose level in diabetic mice and improved the morphological structure and number of pancreatic cells [76]. Hatsumi Ueoka et al. observed an increase in plasma insulin levels after oral administration of both soybean protein isolate solution and soybean peptide solution, with even higher plasma insulin levels in the soybean peptide group at 30 min. This demonstrates a greater insulin secretion-stimulating effect of the peptide due to easier digestion and absorption [74].

Both soybean proteins and peptides have good amino acid sequences and absorption characteristics. However, most research still focuses on the study of the phenotype of diabetic mice, and the underlying mechanisms remain unknown. More in-depth research is still needed. In addition, more refined screening, separation, and purification of peptides with anti-diabetes effects will shed light on the regulation and treatment of diabetes.

### 3.3. The Effect of Soybean Peptides on CVD

CVD is a type of disease involving the heart and blood vessels, including hyperlipidemia, hypercholesterolemia, hypertension, atherosclerosis, and other major diseases. The risk of CVD is closely related to insulin resistance and obesity [103]. At present, CVD accounts for 46.2% of global non-communicable disease deaths, which is one of the main causes of premature death [104]. Nearly 17.9 million people die from CVD each year, and the number of deaths is estimated to increase to 23.6 million by 2030 [105].

The effect of soybean proteins on the improvement and prevention of CVD has received much attention. As early as 1999, the United States Food and Drug Administration (FDA) approved the food label containing soybean proteins to prevent CVD [106]. The FDA approved the health statement that 25 g of soybean proteins per day can reduce the risk of CVD [106]. A large-scale meta-analysis showed that soybean intake was negatively correlated with CVD risk [107]. This may be due to the fact that the soybean bioactive peptides could reduce the total cholesterol levels in the body [108]. Another meta-analysis showed that a daily intake of 25 g of soybean proteins can reduce low-density lipoprotein cholesterol (LDL-C) levels in adults by 3–4% [109].

Both hyperlipidemia and hypercholesterolemia are major risk factors for CVD [110]. Soybean peptides have been reported to lower cholesterol levels. In 2010, scientists first found a new peptide, VAWWMY, with a cholesterol-lowering effect in soybean glycine, named “soystatin,” which has the same binding capacity with bile acid as cholesterol-lowering drugs, although soystatin is the only low-cholesterol peptide isolated from soybean [77]. Three soybean globulin glycin peptides (IAVPTGVA, IAVPGEVA, and LPYP) downregulated the catalytic activity of HMGCoAR, activated the LDLR-SREBP2 pathway, and improved the ability to absorb LDL in vitro, which in turn regulated the cholesterol metabolism of HepG2 cells [78]. In another study, digested soybean protein hydrolysates reduced the solubility of dietary cholesterol micelles by 37.6% and the absorbability by 18.99%, respectively, in Caco-2 cells [79].

Abnormality in blood lipids, especially the elevation of plasma LDL-C levels, is a major risk factor for CVD [80]. In addition to pharmacological methods, researchers are paying more and more attention to nutritional intervention strategies to prevent chronic diseases. In a previous study, two soybean peptides, YVVNPDNDEN and YVVNPDNNEN, both reduced the levels of LDL-C by inhibiting HMG-CoAR activity, while the latter one downregulated the protein level of protein convertase subtilisin/kexin type 9 (PCSK9), a key regulator of LDL-R [80]. In another study, oral administration of two other soybean peptides, ALEPDHRVESEGGL and SLVNNDDDRDSYRLQSGDAL, up-regulated trans-intestinal cholesterol excretion (TICE), inhibited the expression of cytochrome P450 family members (CYP7A1 and CYP8B1), reduced bile acid synthesis, and increased cholesterol excretion in the liver, and thus both peptides have blood lipid-lowering effects [45]. The blood lipid regulatory effect of soybean peptides may play an important role in the treatment of CVD.

Hypertension is also a major risk factor for CVD. Effective treatments for hypertension can reduce the burden of the population with CVD related to high blood pressure [111]. The soybean peptide VHVV inhibited the ACE activity in hypertensive rats and activated the SIRT1-PGC1α/Nrf2 pathway, which reduced the production of renal inflammatory factors and the apoptosis of renal cells. This suggests that VHVV can improve hypertensive renal damage [51]. Another study found that soybean peptides hydrolyzed by alkaline protease and neutral protease had the highest inhibitory effect on ACE activity in hypertensive rats, with an inhibition rate of 71.2% [81]. Ultrasonic fermentation increased the polypeptide content of soybean meal by 36.2%, and thus the ACE inhibitory activity of the soybean meal was increased in vitro by up to 70.05% [82]. In spontaneously hypertensive rats, feeding with soybean oligopeptide at 4.50 g/kg for 30 days significantly reduced both systolic and diastolic blood pressure as well as the quality and concentration of angiotensin II [81]. The investigation of the blood pressure-lowering effect of soybean peptides has mostly focused on the ACE inhibitory activity, and only a very few studies have seen the direct effects of soybean peptides on blood pressure. More studies will be needed to investigate the direct blood pressure-lowering effect in vivo and the effect of anti-hypertension on CVD.

There have been relatively few studies on the effects of soybean peptides on CVD. Relatively few peptides were screened out. Further studies are still needed to screen more potent peptides for CVD-regulating effects and to investigate the underlying mechanisms of the anti-CVD activity of the peptides.

### 3.4. The Effect of Soybean Peptides on Cancer

Cancer has become the second-leading cause of death in the United States [112]. More than 600,000 people in the United States died of cancer in 2021 [112]. Traditional cancer treatments such as drug therapy and chemotherapy are expensive and can cause adverse reactions or complications [113]. In recent years, some anti-tumor peptides have been reported, including soybean peptides [114]. As a relatively inexpensive method, soybean peptides will play an important role in the prevention and remission of cancer development [114].

Lunasin is a bioactive peptide with 43 amino acids and a molecular weight of 5.5 kDa, originally isolated from soybean [83]. It has chemopreventive and therapeutic effects [83]. Studies have shown that Lunasin can effectively inhibit the proliferation of the non-small cell lung cancer (NSCLC) cell line H661 by inhibiting the G1/S phase of the cell cycle and altering the expression of related protein kinase complex components [83]. Thus, the expression level of p27Kip1 and the phosphorylation level of Akt at S473 are altered, and finally, the anti-cancer effect is achieved [83]. Lunasin had a significant inhibitory effect on cell proliferation of human breast cancer cells, and the inhibitory rates of Lunasin extracted from transgenic soybean and wild-type soybean were 43 and 23.8%, respectively [84]. Lunasin at the concentrations of 40 and 80 μM significantly increased the apoptosis of colorectal cancer HCT-116 cells by reducing the level of the DNA repair enzyme (PARP) protein (a marker of cell apoptosis) and increasing the expression of caspase-3 protein and playing a certain role in inhibiting the tumorigenesis by prolonging the G1 phase [85]. As discussed above, Lunasin can inhibit cell proliferation or increase apoptosis in various cancer cells. It could be a potent substance for the prevention and treatment of cancer.

The size of soybean peptides also has a certain influence on the inhibition of cancer cells. Gonzalez-Montoya and colleagues treated three human colon cancer cell lines (Caco-2, HT-29, and HWT-116) with peptides of different lengths obtained after simulating gastrointestinal digestion in vitro. They showed that the inhibitory effects of germinated soybean peptides of different lengths on the proliferation of cancer cells were different [86]. However, more research on the underlying mechanisms of the most active peptides and their potential protective effect on colon health in animal models with colon cancer is still needed. Soybean peptides with different molecular weights inhibited cancer cell proliferation in human blood, breast, and prostate at different degrees, with the 10–50 kDa peptide from the N98-4445A soybean strain inhibiting CCRF-CEM blood cancer cells by 68% [115]. Peptides with different molecular weights from black bean also inhibited the human hepatoma cell line (HepG2), cervical cancer cell line (HeLa), and lung cancer cell line (MCF-7) at different rates, and the inhibitory effects of those with molecular weight < 4 kDa on the growth of HepG2, HeLa, and MCF-7 cancer cells were 2.28-, 1.96-, and 5.91-fold, respectively [47]. The maximum inhibition rate of these peptide segments on the growth of HeLa cancer cells can reach 6.44-fold, which may be related to their hydrophobic interaction and hydrogen bonds with target proteins such as Bcl-2, caspase-7, and caspase-3 [47].

Cancer is a chronic disease that cannot be completely cured at present. The anti-cancer research on soybean peptides is helpful in inhibiting cancer cells. However, only a few studies are focusing on the anti-cancer properties of soybean peptides. More studies will be needed to screen and investigate the anti-cancer effect of more soybean bioactives in animal models with different cancers.

## 4. Conclusions

Chronic diseases are comprehensive diseases with complex pathogenetic mechanisms. With lifestyle changes, chronic diseases have become the main cause of human death in the world. Finding bioactive substances that can regulate and treat chronic diseases has become the unanimous desire of researchers. Bioactive peptides have been more and more recognized for their activities in improving health and preventing or treating chronic diseases. In addition to providing nutrition, food protein peptides can also provide more functions through changes in specific biochemical pathways. Soybean-derived peptides have received a lot of attention for their potent activities of anti-obesity, anti-diabetes, CVD regulation, and anti-cancer, which are very important for the prevention and treatment of chronic diseases (see Figure 4). After digestion, most peptides can be completely absorbed by intestinal cells and transported to the corresponding target organs and cells.

This paper reviewed the bioactivity-related structures of soybean proteins and peptides. It briefly introduced absorption and metabolism in the body and broadly reviewed their functions related to chronic diseases, including anti-obesity, anti-diabetes, CVD regulation, and anti-cancer activity. According to previous reports, soybean proteins and peptides are potent ingredients that may have a major impact on chronic diseases. It is worth further investigation for more potential bioactivities and the underlying mechanisms for these functions.

## 5. Prospect

In the past 20 years, soybean proteins and peptides have attracted extensive attention for their variety of functions. Although it has fewer side effects, it is not as effective as drug therapy. Functional studies on soybean peptides are extensive and inaccurate.

Single-functional soybean peptides have been widely studied, while multifunctional peptides are still a challenging topic. It is necessary to extend the research from single-functional peptides to multifunctional peptides in the future. This change should focus on the method of protein hydrolysis, as protein hydrolysates are complex mixed peptides, and only some of these peptides have bioactive functions.

Although soybean peptides have multiple biological activities, their sensitivity to gastrointestinal proteases and peptidases may lead to a loss of activity before reaching the target organs. This aspect needs to be taken into account, and in vivo or clinical validation will be needed before they can be utilized as a treatment strategy. The application of advanced multi-omics technology and bioinformatics may be of great significance.

The main disadvantage of bioactive peptides is that they are easily degraded by proteases and could be quickly cleared by the kidney, resulting in low bioavailability and poor transmembrane absorption [64]. Small peptides are more easily absorbed than large peptides because they can cross the intestinal barrier more easily and reach their target organs [116]. Soybean peptides have great potential for improving human health. However, there is still a lack of clinical data. In terms of commercialization, how to improve the production technology and their bioavailability while ensuring their quality needs further study.

Generally, food-derived bioactive peptides are characterized by poor absorption, distribution, metabolism, excretion, and toxicity (ADME-T). Currently, ADME characteristics of bioactive peptides have been evaluated in vitro, in vivo, and in silico using various tools [117]. However, how to improve their undesirable characteristics through structural modification or other aspects is still under investigation. The research on improving the bioavailability of soybean peptides will bring great potential and challenges to the development of functional foods or drugs.

Finally, long-term or excessive consumption of soybean peptides may cause allergic reactions and other side effects or reduce digestive capacity in a small group of people. Further research should also investigate the dose and duration for the effective utilization of soybean peptides and avoid potential side effects for the small population. By addressing most of these concerns, soybean peptides will play a significant role in the prevention and treatment of chronic diseases.

## Figures and Tables

**Figure 1 nutrients-15-01811-f001:**
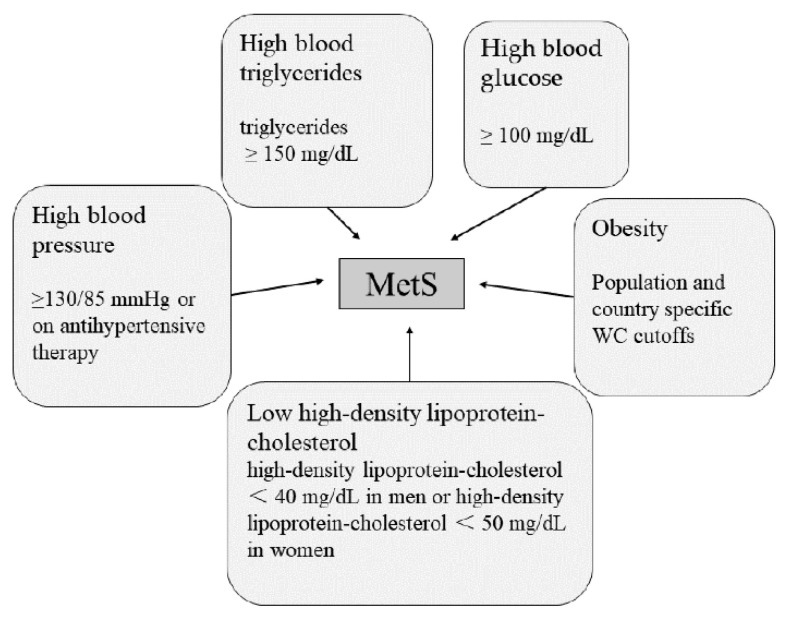
Main diagnosis of MetS. Three of the above five conditions are considered MetS [8].

**Figure 2 nutrients-15-01811-f002:**
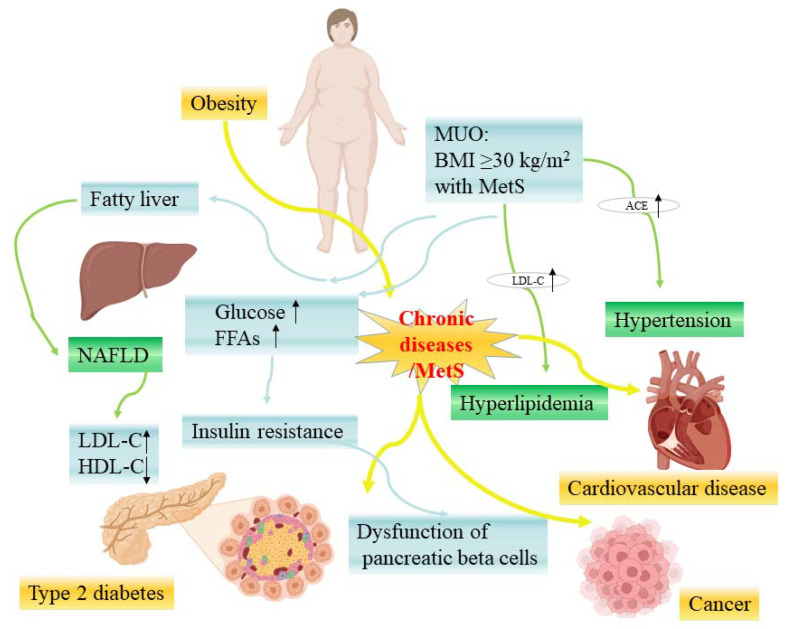
The relationship between chronic diseases and MetS-related diseases (some picture elements are from the BioRender). MUO—metabolic unhealthy obesity; NAFLD—nonalcoholic fatty liver disease; LDL-C—low-density lipoprotein cholesterol; HDL-C—high-density lipoprotein cholesterol; ACE—angiotensin-converting enzyme; FFAs—free fatty acids.

**Figure 3 nutrients-15-01811-f003:**
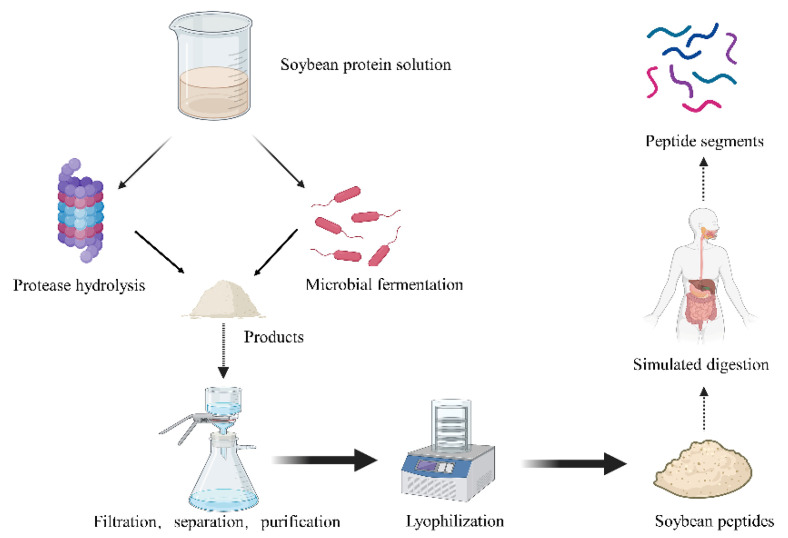
Preparation process of soybean peptide segments (some picture elements are from BioRender).

**Figure 4 nutrients-15-01811-f004:**
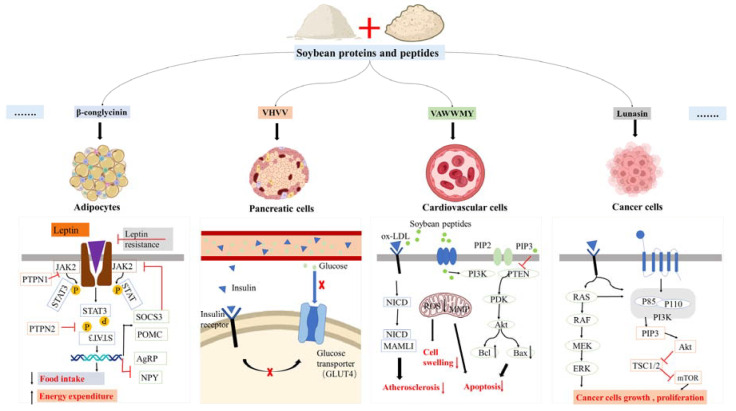
Partial potential mechanisms of the various activities of soybean proteins and peptides on chronic diseases (some picture elements are from the BioRender). JAK—Janus kinase; STAT—Signal transducer and activator of transcription; PTP—protein tyrosine phosphatases; SOCS—suppressor of cytokine signaling; POMC—proopiomelanocortin; AgRP—Agouti-related peptide; NPY—neuropeptide Y; PIP2—phosphatidylinositol-4,5-biphosphate; PIP3—phosphatidylinositol-3,4,5-triphosphate; ox-LDL—oxidized low-density lipoprotein; PI3K—phosphoinositide 3-kinase; PTEN—phosphatase and tensin homolog; ROS—reactive oxygen species; MMP—matrix metalloproteinase; NICD—Notch intracellular domain; MAML1—Mastermind-like proteins; MEK—mitogen-activated protein kinase; ERK—extracellular signal-regulated kinases; mTOR—mammalian target of rapamycin.

**Table 1 nutrients-15-01811-t001:** The functions of soybean proteins and peptides in relation to chronic diseases.

Function	Bioactive Substances of Soybean Peptides	Detection Model (Females or Males)	Main Results	References
Anti-obesity effect	β-conglycinin	C57BL/6 mice (males)	Weight decreased.	[66]
	β-conglycinin	C57BL/6 mice (males)	FGF21 increased.	[67]
	β-conglycinin	Obese rats (males)	Abdominal fat and lipid contents decreased.	[68]
	β-conglycinin	Rats (males)	Serum cholesterol decreased from 146 mg/dL to 124 mg/dL, and liver triglycerides decreased from 214 mg to 163 mg.	[69]
	Soybean protein isolates	Obese rats (females)	AST level decreased from 222.5 U/L to 103.4 U/L, and ALT level decreased from 71.9 U/L to 56.2 U/L.	[70]
	Soybean proteins	Obese OLETF rats (males)	Serum cholesterol decreased to 142 mg/dL.	[71]
	Soybean proteins	C57BL/6J mice (males)	*Firmicutes* to *Bacteriodetes* increased; Serum triglycerides decreased.	[72]
Anti-diabetes effect	Glu-Ala-Lys and Gly-Ser-Arg		The inhibitory effect of α-glucosidase activity was 45.89%.	[73]
	Soybean protein isolates and soybean peptides	Human (both)	Plasma insulin response significantly increased after 30 min of SPI consumption.	[74]
	Soybean proteins	Patients with diabetes (both)	Fasting blood glucose decreased by 1.68% after 2 months.	[75]
	VHVV	H9c2 cells and ICR mice (males)	Cell viability increased; Cell apoptosis decreased; Postprandial blood glucose level decreased.	[76]
Anti-CVD	VAWWMY/ Soystatin	Rats (males)	Serum and liver cholesterol levels were reduced to 0.03%.	[77]
	IAVPTGVA, IAVPGEVA and LPYP	HepG2 cells	Catalytic activity of HMGCoAR and the level of LDL decreased.	[78]
	Soybean protein hydrolysates	Caco-2 cells	The solubility of dietary cholesterol micelles decreased.	[79]
	YVVNPDNDEN and YVVNPDNNEN	HepG2 cells	After 24 h, the relative expression of LDL-C and PCSK9 decreased by about 20%.	[80]
	ALEPDHRVESEGGL and SLVNNDDDRDSYRLQSGDAL	Caco-2 cells	Blood lipids decreased.	[45]
	VHVV	Hypertensive rats (males)	ACE activity and inflammatory factors decreased.	[51]
	Small molecule peptides	Hypertensive rats (males)	The inhibition rate of ACE activity was about 60% and the concentration of angiotensin II decreased.	[81]
	Polypeptide content of soybean meal		ACE activity decreased.	[82]
Anti-cancer effect	Lunasin	NSCLC cell line H661	Decreased proliferation of cancer cells.	[83]
	Lunasin	Human breast cancer cells	Decreased proliferation of cancer cells.	[84]
	Lunasin	Colorectal cancer HCT-116 cells	After treatment with 10 µM lunasin for 72 h, cell growth decreased by 12.9%.	[85]
	Germinated soybean peptides	Human colon cancer cell lines	After treatment with 10 mg/mL soybean peptide segments for 24 h, cell viability decreased by 82–66%.	[86]
	Black soybean peptides Leu/Ile-Val-Pro-Lys	HepG2 cells MCF-7cells HeLa cells	With high cytotoxicity, the IC_50_ values are 0.22, 0.15, and 0.32 µM, respectively.	[47]

Note: FGF21: fibroblast growth factor 21 gene; AST: aspartate aminotransferase; ALT: alanine aminotransferase; SPI: soybean protein isolates; HMGCoAR: 3-hydroxy-3-methylglutamate CoA reductase; LDL-C: low-density lipoprotein cholesterol; PCSK9: protein convertase subtilisin/kexin type 9; ACE: angiotensin-converting enzyme; IC_50_: half maximal inhibitory concentration.

## Data Availability

No new data were created or analyzed in this study.

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
