# Peer review of "The Beneficial Effects of Soybean Proteins and Peptides on Chronic Diseases"

_nutrients, 2023, doi:10.3390/nu15081811_

Round 1

Reviewer 1 Report

//

This review by Sumei Hu et al named “The beneficial effects of soybean proteins and peptides on chronic diseases” is a very well-written comprehensive study. It deeply investigates the role of the different soybean peptides in animal studies. It also recognizes that there is lack of literature concerning the effects of dietary consumption of soybean in humans and the need to further investigate this.

My main concern is that this paper does not address if the effects of these peptides can differentially affect females and males. It could be of interest to specify this in the text and in the tittle if these studies were done in males, females, or both.

Reviewer 2 Report

Manuscript: The beneficial effects of soybean proteins and peptides on chronic diseases.

The manuscript by Hu et al. briefly presented the literature updates on soybean-derived peptides and proteins in health, especially chronic diseases. Overall, the manuscript is interesting, and requires significant revision as follows:

Comments

1.       The abbreviations can be cross-verified separately in the abstract and main text.

2.       Line 27, please add information on the mechanism of chronic diseases manifestation such as “In chronic diseased conditions, however, the protease network imbalance may directly influence the critical regulatory system, resulting in a persistent malfunction of the complex cellular signaling network.” i.e. https://doi.org/10.3390/biomedicines10102477.

3.       Section 3.1 and 3.4,  in cases of obesity and cancer, the microbial population inherited as microbiota (diverse microbial populations) also plays a crucial role. However, the interaction among the microbial population (via signaling molecules/communication) is vital for good health via their preservation through diet/supplements. Please add such information in these sections to highlight the significance, i.e, https://doi.org/10.3390/molecules27217584

4.      The literature descriptions in each section can be minor polished with quantitative results.

5.      The illustration quality should be significantly improved.

6.      Figure 4 can be more elaborated with the detailed mechanisms.

7.      Table 1, please add quantitative information/descriptive results as significance.

8.      Citations should be updated (some) as recent within 2022-2023.

Reviewer 3 Report

The manuscript from Hu and collaborators is a review of the beneficial effects of soybean proteins and peptides on different types of chronic diseases. 

Overall, the authors did a great job in summarizing the studies reporting the effects of soybean 
proteins and peptides. They presented the perspectives of the soybean proteins and peptides, including the limitations for therapeutic uses.

However, some few adjustments are needed before the fanlike acceptance.

For instance, in some parts the authors should make clear what compound is involved in the respective action they are describing. The information if the effect was observed in human or animal must also be indicated. 

The authors should also attempt to the use of italics for Latin words such as in vitro and in vivo.

The quality of the figures should be improved.

Other comments are provided in the attached pdf file.

Round 2

Reviewer 2 Report

Accept